hypertension; precision medicine; genomics

**Corresponding author:**
Patricia B. Munroe;
Email: p.b.munroe@qmul.ac.uk

# Harnessing the power of genomics in hypertension: tip of the iceberg?

Hafiz Naderi[1,2,3] [ID], Helen R. Warren[1,3] [ID] and Patricia B. Munroe[1,3] [ID]

[1]William Harvey Research Institute, Queen Mary University of London, London, UK; [2]Barts Heart Centre, St Bartholomew's Hospital, West Smithfield, London, UK and [3]National Institute of Health and Care Research Barts Biomedical Research Centre, Queen Mary University of London, London, UK

## Abstract

Despite the blaze of advancing knowledge on its complex genetic architecture, hypertension remains an elusive condition. Genetic studies of blood pressure have yielded bitter-sweet results thus far with the identification of more than 2,000 genetic loci, though the candidate causal genes and biological pathways remain largely unknown. The era of big data and sophisticated statistical tools has propelled insights into pathophysiology and causal inferences. However, new genetic risk tools for hypertension are the tip of the iceberg, and applications of genomic technology are likely to proliferate. We review the genomics of hypertension, exploring the significant milestones in our current understanding of this condition and the progress towards personalised treatment and management for hypertension.

## Impact statement

High blood pressure or hypertension is the strongest modifiable risk factor for cardiovascular disease, and it is a complex condition influenced by both genetic and environmental factors. In this review article, we explore the significant milestones to our current understanding of the genetics of hypertension. We highlight key landmarks in blood pressure-related research from the discovery of monogenic forms of hypertension to the era of genome-wide association studies. Alongside the development of polygenic risk scores for cardiovascular risk prediction, the application of multi-omics, single-cell RNA technologies and machine learning are providing new insights into the pathophysiology of hypertension. We also explore the development of pharmacogenomics in hypertension and the role of large-scale biobanks in drug development together with the challenges and future landscape.

## The hype in hypertension

Hypertension is the strongest modifiable risk factor for cardiovascular disease, being responsible for the majority of stroke and up to half of cases of coronary heart disease (Perkovic et al., 2007). The global health burden of hypertension is immense, with 1.5 billion people projected to be affected by 2025 (Kearney et al., 2005). The societal cost resulting from the morbidity and mortality caused by hypertension has raised an urgent need for innovative approaches, with its prevention being a top priority in governments worldwide and endorsed by the 2023 European Society of Hypertension guidelines (Mancia et al., 2023) and the latest 2024 European Society of Cardiology guidelines (McEvoy et al., 2024). The risk for cardiovascular disease attributable to blood pressure (BP) is on a continuous exposure scale (Murray et al., 2020). Elevated BP adversely affects the heart, kidneys, brain, eyes and vessels, leading to structural and functional changes termed hypertension-mediated organ damage. Clinically, hypertension is diagnosed based on BP measurements, however, BP is a complex trait influenced by a magnitude of physiological and environmental interacting pathways. Approximately 95% of cases of hypertension are referred to as primary or essential hypertension (EH) with genetics contributing approximately 30% of BP variance, and the remainder due to lifestyle factors (Poulter et al., 2015). The other 5% of causes is termed secondary hypertension, of which 1% are monogenic disorders (Cowley, 2006). Currently, pharmacological treatments for hypertension are introduced when BP measurements are elevated and there is potential end-organ damage initiation. The major international guidelines recommend a combination of antihypertensive drugs as first-line therapy to improve efficacy and reduce the risk of side effects related to treatment. Challenges such as non-adherence to therapy and resistant hypertension have limited the current ability to ensure adequate BP control in the general population. The 'precision





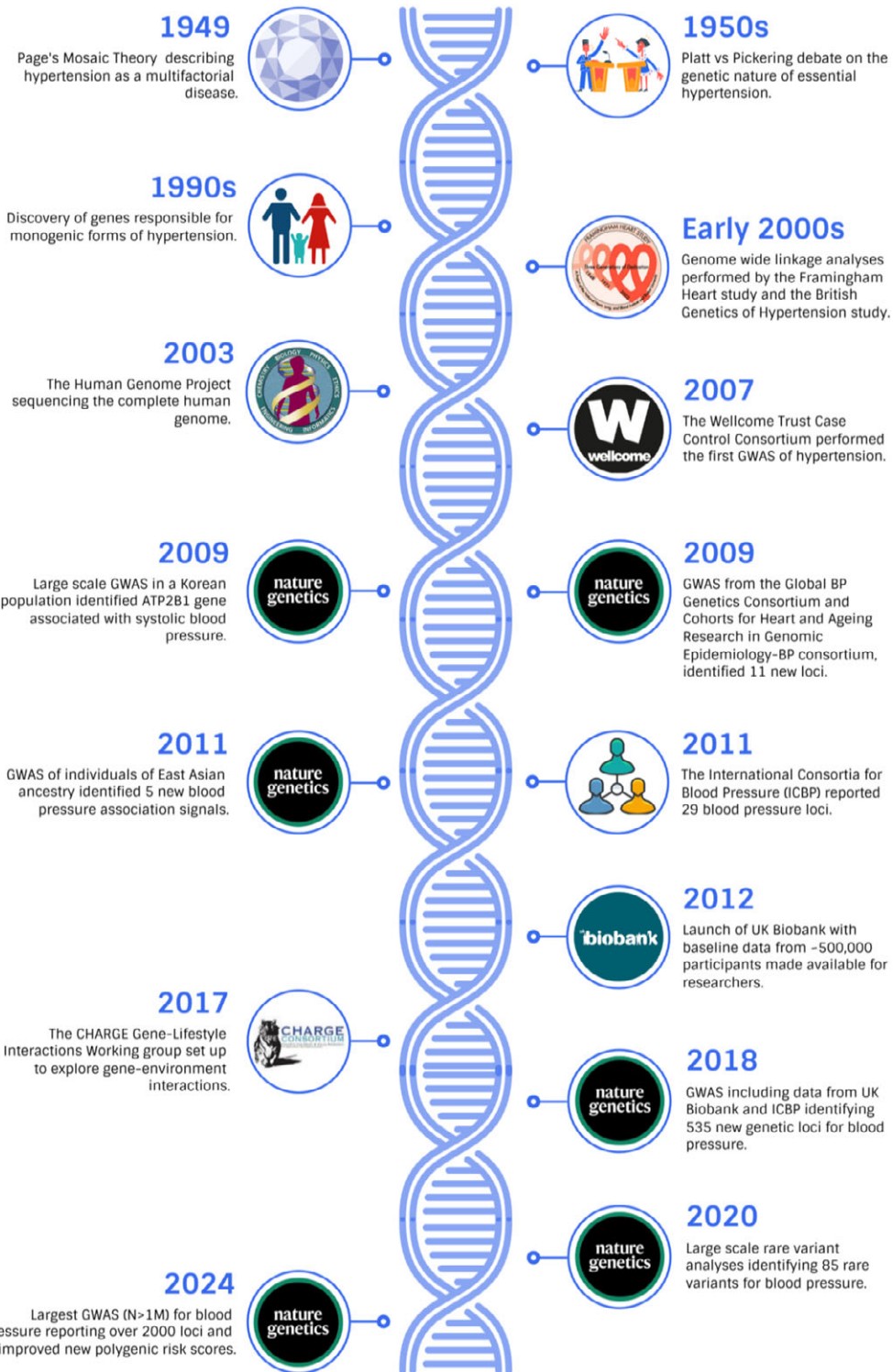

**Figure 1.** Key advances in hypertension genomics.

hypertension' approach has been proposed to consider an individual's unique characteristics for better-targeted risk profiling and treatment strategy (Dzau and Hodgkinson, 2024). In this review, we highlight some of the key advances (Figure 1) in hypertension genomics together with its challenges and future landscape.

## The yellow brick road

The concept that hypertension is a multifactorial disease was first proposed by Page's Mosaic Theory in 1949 (Page, 1949). Following evidence from familial studies and the discovery of rare monogenic disorders of hypertension, the genetic contribution of hypertension was recognised. The 1950s witnessed the legendary Platt *vs.* Pickering debate on the genetic nature of EH. The controversy stemmed from the appearance of BP frequency distribution curves, which led to the discussion of its monogenic or polygenic potential. Platt regarded EH as a distinct condition with rare variants of hypertension as evidence for its single-gene inheritance, whilst Pickering postulated that hypertension was seen only in the extreme of a continuous distribution curve of BP values and therefore determined by a collection of genes (Zanchetti, 1986; Brown, 2012). Later studies supported the polygenic theory of hypertension with a wide range of heritability estimates for systolic and diastolic BP from 6% to 68% (Kolifarhood et al., 2019). Differences in environmental conditions, type of study design, trait definition, and analytical techniques may explain the wide variation in heritability estimation of BP traits. In light of the new evidence, Page also acknowledged the genetic influence of hypertension in his revision of the Mosaic Theory in 1982 (Page, 1982). The 1990s saw a big boom in gene mapping with the launch of The Human Genome Project sequencing the complete human genome by 2003 (International Human Genome Sequencing Consortium, 2004). During this period there were a number of genome-wide linkage analyses performed, including the Framingham Heart Study and the British Genetics of Hypertension (BRIGHT) study, identifying regions in the DNA linked to variations in BP (Levy et al., 2000; Caulfield et al., 2003). Although an important step, interpreting results from linkage analyses proved challenging, as the regions in the DNA identified were large, hence difficult to identify the responsible gene. The turn of the millennium welcomed a wave of advancing technology and bioinformatics, paving the way for the era of genome-wide association studies (GWAS).

## GWAS for blood pressure

The hunt for genes implicated in BP regulation has been challenging. Before the advent of GWAS, genes and mechanisms for BP were mostly discovered using rat and mouse models and candidate gene studies (Lerman et al., 2019). Since the launch of single-nucleotide polymorphism (SNP) genotyping arrays in 2005, BP-GWAS of increasing scale has been performed. The first GWAS of hypertension was performed in 2007 by The Wellcome Trust Case Control Consortium (WTCCC). This consortium undertook GWAS of 2,000 cases and 3,000 shared controls for seven complex diseases, including hypertension (Wellcome Trust Case Control Consortium, 2007). Although no single SNP achieved genome-wide significance ($p < 5 \times 10^{-7}$), six variants were found to have suggestive associations with hypertension ($p < 5 \times 10^{-5}$). The Family Blood Pressure Program (FBPP) subsequently focussed on these six SNPs in a study of 11,433 individuals recruited from hypertensive families. This study did not replicate the results of the WTCCC study, however, one of the six SNPs (rs1937506) was found to be associated with hypertension in Hispanic Americans and European Americans (Ehret et al., 2008). Subsequently, investigators from the Korean Association Resource (KARE) project analysed the association of the six SNPs in 7,551 unrelated individuals in Korea. The authors reported one intronic SNP (rs7961152 at the *BACT1* gene locus) to be associated with hypertension risk (odds ratio 1.29, 95% confidence interval 1.01–1.64, $p = 0.004$) (Hong et al., 2009). The WTCCC, FBPP and KARE studies demonstrated that due to the complex genetic architecture of hypertension, a larger sample population may be necessary to identify genetic variants implicated in BP. To increase the sample sizes, consortia were established, to combine data together across many different studies. The first exciting results in BP-GWAS were in 2009 from large meta-analyses of GWAS ($n = 34,433$) from the Global BP Genetics Consortium (GBPGEN) and Cohorts for Heart and Ageing Research in Genomic Epidemiology-BP (CHARGE-BP) consortium, identifying 11 new loci (Levy et al., 2009; Newton-Cheh et al., 2009; Psaty et al., 2009). Seven of these loci were also subsequently reported in a Japanese population (Takeuchi et al., 2010). Following this success story, the two consortia GBPGEN and CHARGE-BP merged to form the International Consortia for BP (ICBP) identifying more novel loci in 2011 (Wain et al., 2011; Ehret et al., 2011). Cho et al., (2009) also reported 5 new BP loci in East Asians (Cho et al., 2009). The results from these studies provided new insights into the biology of BP with opportunities for developing new therapies.

## Big data, biobanks and beyond

Over the past decade, large-scale datasets have been developed, one example is the UK Biobank, permitting analysis of up to 500,000 richly phenotyped participants (Bycroft et al., 2018). Leveraging these resources, Warren et al. (2017) performed the first UK Biobank BP-GWAS for the first 150,000 genotyped participants (Warren et al., 2017). Hoffman et al. (2017) performed a GWAS on long-term average BP from the electronic health records of 99,785 individuals identifying 39 new loci (Hoffmann et al., 2017). Once all the UK Biobank data became available, Evangelou et al., (2018) performed GWAS meta-analyses including data from both the UK Biobank and ICBP and identified 535 new genetic loci influencing BP (Evangelou et al., 2018). Alongside the GWAS for common variants for BP, there have also been large-scale consortium-based studies focused on the discovery of rare variants across many meta-analysed studies, including UK Biobank (minor allele frequencies of <1%) (Surendran et al., 2020). These studies have yielded >80 rare variants, all having larger effects on BP (~1.5 mmHg per allele, compared to ~0.5 mmHg for common variants) (He et al., 2022).

Additional large biobanks include the Million Veteran Program (MVP, $n$ ~ currently recruiting and with 635,969), which has created one of the largest epidemiologic research infrastructures embedded within the national health care system operated by the US Department of Veteran Affairs, these data have also been used for BP-GWAS (Giri et al., 2019; Verma et al., 2024). Other notable cohorts projected to deliver population-level genomic insights include: Genomics England's first initiative, the 100,000 Genomes Project identifying the genetic causes of many rare diseases; and FinnGen, a Finnish biobank of 500,000 participants (100,000 Genomes Project Pilot Investigators et al., 2021; Kurki et al., 2023).

The majority of GWAS studies for BP did not initially consider the precise role and biological significance of gene–environment interactions (GxE). To address this gap in knowledge, the CHARGE Gene-Lifestyle Interactions Working Group was formed, and this group has conducted a series of genome-wide interaction studies for various traits and exposures. Recently, the group examined interactions between genotype and the Dietary Approaches to Stop Hypertension (DASH) diet score and systolic BP (Guirette et al., 2024). They demonstrated gene-DASH diet

score interaction effects on systolic BP in several loci in European population-specific and cross-population meta-analyses. Additional studies have investigated several other important lifestyle factors, including a study investigating BP × Alcohol, which found 54 loci; and BP × Smoking which found 15 loci (Sung et al., 2018; Feitosa et al., 2018). These studies included ~130,000 individuals across multi-ancestry data-sets, but there were limited findings, and analyses in larger sample sizes are currently ongoing.

## Polygenic risk score and cardiovascular risk prediction

As GWAS results become publicly available, this has enabled risk prediction modelling to include genetic biomarkers for clinical applications. BP is a highly polygenic trait, influenced by thousands of different SNPs each of which has a small effect on BP. Polygenic risk scores (PRS) have been developed by combining the risk associated with many common DNA sequence variants into one single aggregated risk score (Lewis and Vassos, 2020). The first genetic risk score for BP was developed by the ICBP in 2011 by combining together 29 different significant genetic variants into one score (Ehret et al., 2011). The identification of further BP loci has led to the development of PRS with increasing performance to estimate an individual's risk of hypertension. For example, in 2022, Parcha et al., developed and tested a BP-PRS in a multi-ancestry US cohort ($n = 21,897$) to evaluate the relative contributions of the traditional cardiovascular risk factors to the development of adverse events in the context of varying BP risk profiles in individuals with no previous cardiovascular disease. They demonstrated that the PRS had an incremental value beyond traditional risk factors highlighting the potential of incorporating genetic information into risk estimates (Parcha et al., 2022). Recently, Keaton et al., (2024) performed the largest single-stage BP GWAS to date ($n = 1,028,980$ European ancestry individuals), reporting a total of 2,103 independent genetic signals for BP. The BP-PRS generated from this study revealed clinically meaningful differences in BP (16.9 mmHg systolic BP, 95% CI = 15.5–18.2 mmHg, P = $2.22 \times 10^{-126}$) and more than a seven-fold higher odds of hypertension risk (OR = 7.33; 95% CI = 5.54–9.70; P = $4.13 \times 10^{-44}$), when comparing individuals in the top (highest genetic risk) *versus* bottom (lowest risk group) deciles of the PRS in an independent European cohort, Lifelines. The authors also showed that the BP-PRS was significantly associated with higher BP in individuals of African-American ancestry from the All-of-Us Research program in the United States (Keaton et al., 2024).

As part of the study design for the large meta-analyses for BP-GWAS, the impact of biological sex has been understudied, thus results are limited in assessing differences between sexes. Kauko et al. (2021) developed a sex-specific PRS in FinnGen ($N = 218,792$) and found the female PRS was more strongly associated with hypertension in women than the male PRS in men (Kauko et al., 2021). Similarly, Shetty et al. (2023) developed sex-specific systolic BP-PRS in UK Biobank and tested for associations of developing hypertension in 212,669 participants in the All of Us study. They found the genetic risk of systolic BP was more strongly associated with female PRS (Shetty et al., 2023). Recently, Yang et al. (2024) performed sex-stratified GWAS analyses of BP traits in the UK Biobank resource, identifying 1,346 previously reported and 29 new BP trait-associated loci. Despite equal sample sizes, sex-stratified GWAS of systolic BP, diastolic BP and pulse pressure identified 1.8-fold more loci in the female-only analyses ($N = 174,664$) than in the male-only analyses ($N = 174,664$). These sex-specific loci were enriched for hormone-related transcription factors, in particular, oestrogen receptor 1, and sex-specific polygenic association of BP traits was associated with multiple cardiovascular traits (Yang et al., 2024).

Integration of PRS for early disease risk prediction is an area of active research, and with an increased number of loci being found for complex diseases, the percentage of the heritability explained is increasing, and better PRS are being developed (Ge et al., 2019). With increasing datasets being recruited of non-European ancestry, new loci discovery and population-specific PRS are being developed (Fujii et al., 2024). Genomics PLC have sought to integrate PRS to re-engineer prevention strategies in healthcare for commercial exploitation (Genomics PLC). In their trial, Fuat et al., (2024) enrolled 832 participants across 12 UK primary care practices. They observed that the integration of genetic data to a conventional risk algorithm (QRISK2) for cardiovascular disease was accepted by healthcare professionals and participants in primary care with planned changes in prevention strategies (Fuat et al., 2024). These risk prediction tools are a funnel for personalised medicine with potential for population-level risk stratification. However, currently, it is unclear how this genetic information is best integrated into guideline-recommended risk prediction tools. One of the main weaknesses of PRS is that they report genetic risk relative to a given population, thus their contribution is only meaningful in the context of other risk factors, limiting their clinical applicability (Ding et al., 2021; Abramowitz et al., 2024).

GWAS provides candidate genes, disease mechanisms and PRS for assessing relationships between BP and other traits. The PRS however do not provide information on whether there are causal relationships. Mendelian randomisation (MR) is being widely applied to infer causality using genetic data, with power equivalent to that of a randomized controlled trial, overcoming traditional bias attributed to confounders and reverse causation (Burgess et al., 2012). There have been several applications of BP in an MR framework (Nazarzadeh et al., 2019; Tang et al., 2023). In an MR study by Clarke et al., (2023), higher levels of genetically predicted systolic BP were associated with higher risks of major cardiovascular disease in the range of 120 to 170 mmHg of participants in the China Kadoorie Biobank (Clarke et al., 2023). The associations of lower genetically-predicted systolic BP with lower risks of cardiovascular outcomes down to 120 mmHg challenge the conventional strategy of restricting the initiation of BP-lowering medication to people with systolic BP ≥140 mmHg. These findings provide support for lowering systolic BP for a wider range of the population down to 120 mmHg.

## From omics to AI for gene identification

Advances in multi-omics technologies have provided new insights into the pathophysiology of hypertension. The omics approaches target different molecular levels, including the genome, transcriptome, proteome, metabolome and microbiome, providing a comprehensive assessment of the processes by which DNA is transcribed into RNA that is translated into proteins that regulate downstream metabolism. These novel datasets can provide valuable insights into the mechanisms of hypertension, allowing for a better understanding of its pathogenesis and aiding the clinical needs of early diagnosis and monitoring of the treatment response. Several computational approaches have been used to prioritise candidate genes leveraging multi-omic datasets, with most groups using the

GWAS results from the 2018 Evangelou et al., study (Evangelou et al., 2018). For example, Eales et al. (2021) integrated genotype, gene expression, alternative splicing and DNA methylation profiles of up to 430 human kidneys to characterise the effects of BP SNPs from GWAS on renal transcriptome and epigenome (Eales et al., 2021). Sheng et al. created maps of expression quantitative trait loci (eQTLs) for 659 kidney samples and identified cell-type eQTLs, and integrated GWAS results with single-cell RNA sequencing (scRNA-seq) and a single-nucleus assay for transposase-accessible chromatin with high-throughput sequencing, a method for identifying regulatory elements in specific cell types. Their study indicated 200 genes for kidney function and hypertension and highlighted endothelial cells and distal tubules as being important for BP (Sheng et al., 2021). More recently, Ganji-Arjenaki et al. (2024) leveraged the largest GWAS of BP traits with scRNA-seq from 14 mature human kidneys and prioritised myofibroblasts and endothelial cells among the 33 annotated cell types involved in BP regulation (Ganji-Arjenaki et al., 2024). Other efforts include van Duijvenboden et al. (2022) who conducted annotation-informed fine-mapping incorporating tissue-specific chromatin segmentation and colocalisation using transcriptomics and additional gene prioritisation utilising both scRNA-seq and proteomics datasets to identify causal variants and candidate effector genes for BP traits (Duijvenboden et al., 2023). Kamali et al. (2022) also developed a pipeline to leverage epigenomic and transcriptomic datasets and identified 1,880 prioritised genes for BP and downstream from this, the genes were assessed for druggability and tested for functional enrichment (Kamali et al., 2022). The different approaches have highlighted many BP genes for follow-up studies.

Machine learning (ML) approaches have also been used to prioritise candidate genes discovered through GWAS. ML algorithms build mathematical models that are learnt from training data to make predictions. ML in GWAS has been used to boost statistical power of GWAS, refine PRS produced from GWAS and prioritise candidate genes in post-GWAS-analysis (Li et al., 2017; Nicholls et al., 2020). Additionally, multi-parallel functional experiments have also been applied to gain insights into causality and related molecular mechanisms of genetic variants derived from GWAS. Oliveros et al. (2023) functionally characterised 4,608 genetic variants in linkage with SNPs at 135 BP loci in vascular smooth muscle cells and cardiomyocytes using parallel reporter assays. This approach demonstrated the potential to identify functionally relevant variants for a better understanding of BP genetic architecture (Oliveros et al., 2023).

## Pharmacogenomics, therapeutics and druggability

Genetics research has promoted the discipline of pharmacogenomics exploring the influence of genomic variation on an individual's response to BP therapy (Roden et al., 2019). This is particularly necessary for hypertension as there is a large proportion of individuals who do not respond to current treatments. The development of new drug treatments is therefore one key driver of BP genomics and exploring potential for drug repurposing. Evangelou et al. (2018) discovered five loci containing genes that are drug targets for several known antihypertensive classes and Surendran et al. (2020) reported 23 genes as potential drug targets (Evangelou et al., 2018; Surendran et al., 2020). Similarly, Keaton et al. (2024) used transcriptome-wide association studies (TWAS) to identify 38 genes, including an established drug target for BP medications (ADRA1A) and five genes targeted by other approved drugs

(Keaton et al., 2024). However, as previously described, GWAS and downstream bioinformatics analyses do not pinpoint the causal gene, they only provide candidates for further exploration. Functional cellular studies and the development of animal models remain important tools once a gene is identified as having strong potential as a druggable target.

Drug-gene interaction databases have enabled a comprehensive catalogue of druggable genes (Gaulton et al., 2017; Cotto et al., 2018). These open-access online resources have allowed a search by gene of drug-gene interactions or potential for druggability. Canagliflozin, an SGLT2 inhibitor, is an approved and widely used medication in the treatment of type 2 diabetes targeting the gene SLC5A1. However, it was noted that it reduced systolic BP in individuals with type 2 diabetes and chronic kidney disease, providing end-organ protection for this cohort of patients who experience a high burden of hypertension (Ye et al., 2021). Although it is currently not licenced for BP treatment, it highlights the repurposing potential of existing drugs.

The most common distinct cause of hypertension is primary hyperaldosteronism, also known as Conn's syndrome. It has been shown that some patients with treatment-resistant hypertension, defined as uncontrolled, high BP despite being on three or more different antihypertensive drug classes, have increased aldosterone production. Baxdrostat, an aldosterone synthase inhibitor, targets the gene CYP11B2, which encodes aldosterone synthase in the adrenal gland. The CYP11B2 candidate gene was found to be genome-wide significant in BP-GWAS of Japanese individuals by Kanai et al. (2018) and also in subsequent European ancestry BP-GWAS (Keaton et al., 2024). It is a once daily oral medication currently under study with promising phase 2 clinical trial results, which may expand the possible choices of therapeutic agents for treatment-resistant hypertension (Freeman et al., 2022).

The biological architecture of hypertension is complex, and existing medications target only specific mechanisms in BP regulation, with variable effectiveness across individuals (Thomopoulos et al., 2015). The development of gene-editing and RNA-based approaches has inspired new treatment modalities for hypertension. These techniques allow selective and organ-specific modulation of systems involved in BP regulation. Antisense oligonucleotides (ASO) and small interfering RNA (siRNA) have been used to specifically target the hepatic angiotensinogen (AGT) production, with the scope of effectively downregulating the activation of the renin-angiotensin system (Masi et al., 2024). These approaches have the potential to simplify BP treatment regimens with weekly, monthly or even once-only injection of the drugs. Among the various technologies, siRNA and ASO that reduce hepatic AGT production are currently in advanced development, with phase I and II clinical trials showing their safety and effectiveness (Desai et al., 2023; Bakris et al., 2024). The CRISPR (clustered Regularly Interspaced Short Palindromic Repeats) and its associated protein Cas9 is another gene editing tool and the first CRISPR-based human therapy was approved in 2023 for sickle cell disease and β-thalassaemia (Wong, 2023). CRISPR-Cas9 gene editing technology has also been utilised in hypertension research in animal models (Cheng et al., 2017; Sun et al., 2021). The application of gene-editing may be an avenue for treating single-gene causes of hypertension. Examples of monogenic hypertension include Liddle syndrome (epithelial sodium channel gain of function), Gordon syndrome (gain of function in 4 genes regulating Na-K-Cl cotransporter activity), mineralocorticoid excess (11-β-hydroxysteroid dehydrogenase type II loss of function), and glucocorticoid-remediable aldosteronism (crossover of adjacent genes, CYP11B1 and CYP11B2 as

previously mentioned) (Zappa et al., 2024). Monogenic forms of hypertension are typically associated with early onset, severe, and resistant hypertension. In cases of monogenic hypertension, where a single gene mutation follows Mendelian inheritance patterns, gene-editing may offer a cure to the disease.

## Challenges and future landscape

High-throughput next-generation sequencing technologies continue to evolve, and the cost of whole genome sequencing is continuing to fall. This will increase datasets for dissection of causal genes, BP mechanisms and data for inclusion in risk score algorithms. There are many different ethical issues raised by genomic research. Ancestral bias is an important consideration as most genetic data acquired to date has been predominantly from individuals of European ancestry. Individuals of African ancestry have the highest age-adjusted prevalence of hypertension but are relatively under-represented in BP genetic studies (Franceschini et al., 2013). An increase in diverse sampling is being addressed by ongoing efforts of national biobanks which are being used to discover novel and ancestry-specific loci within, for example, Japan, Asia, Africa and Qatar (Genome Research Biobank Project Biobank Japan; GenomeAsia; H3Africa – Human Heredity & Health in Africa; Qatar Biobank). Despite these efforts to increase genetic diversity and representation, there is still more to be done. Bridging this data gap is crucial for equitable genomic testing and ensuring GWAS results are beneficial across populations, and that we avoid reinforcing existing health disparities. Furthermore, genomic data can reveal sensitive information about an individual and their family's ancestry and health. It is therefore important that biobanks store and provide approved researchers with access to genomic data securely and responsibly. There are also ethical considerations in incorporating genetic risk stratification with implications in the insurance sector. An important step in implementing ethical and governance frameworks that balance these risks will be to ensure that any procedures command public trust.

Demographic changes with the ageing population and increased multimorbidity pose challenges to hypertension management. To date, genomic research in hypertension has been largely focused on aiding diagnosis (especially for monogenic forms of hypertension) and identifying potential target mechanisms for treatment. The next wave of genomics in hypertension has the potential to empower a preventive approach with effective screening at a population scale. Our Future Health, the flagship UK programme with the National Health Service, highlights a strategic partnership between industry, academia, and government together with patients and the public to support preventive approaches to tackling common diseases (Our Future Health). The programme combines clinical and genetic data to calculate disease risk scores with the aim of targeting individuals who are at higher risk of developing certain diseases. This will provide an opportunity to test the potential of new polygenic risk scores in health care and of new diagnostic tests or treatments to see how effective they could be for people at higher risk of certain diseases. These collaborations demonstrate that the long-term applications of genomic technology are likely to proliferate beyond genetic risk tools. New use cases are appearing and will revolutionise healthcare delivery through improved differential diagnosis with genetics and personalised medication selection, optimising safety and efficacy. These hold promise for implementing predictive, personalised and preventive approaches to hypertension management that is enduring. Central to the

effective delivery of precision medicine in hypertension is patient and public involvement. A focus on a person-centred approach with emphasis on the patient perspective in research, guidelines and scientific documents ensures that the patient is at the heart of all that we do.

**Open peer review.**   To view the open peer review materials for this article, please visit http://doi.org/10.1017/pcm.2025.1.

**Acknowledgements.**   Graphical abstract was created using Procreate.com.

**Author contribution.**   HN wrote the manuscript. HRW and PBM critically reviewed and approved the manuscript.

**Financial support.**   HN acknowledges the National Institute for Health and Care Research Integrated Academic Training Programme, which supports his Academic Clinical Lectureship post (CL-2024-2119-002). PBM and HRW acknowledge support from the National Institute for Health and Care Research Biomedical Research Centre at Barts (NIHR202330).

**Competing interest.**   The authors declare no competing interests exist.

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
