## [Reviewer Report]

This manuscript focuses on genomic research on hypertension and discusses the historical development, current progress and future prospects of this field. But there are still some problems.

Major comments:

a) It is suggested that an additional paragraph be added to the Introduction section to provide more specific information on the clinical importance of hypertension, the limitations of existing treatments, and the potential role of genomics in addressing these issues.

b) Consider adding a timeline chart that visually illustrates the key milestones from the Platt-Pickering debate in the 1950s to modern large-scale GWAS studies.

c) The manuscript mentioned a variety of research methods, such as GWAS, PRS, etc. A section could be added to discuss the advantages and disadvantages of these methods and their specific applications in hypertension research.

d) To explore the advantages of GWAS in discovering new gene loci and its limitations in explaining causal relationships.

e) Expand the scope of multi-omics research and discuss how multi-omics integration can help us understand the molecular mechanisms of hypertension more comprehensively.

f) The manuscript did not explore sex differences in hypertension genomics research.

g) The manuscript mainly focuses on genetic factors, but there is less analysis of the interaction between genes and environmental factors (such as diet, stress, lifestyle, etc.).

h) Single-cell sequencing technology applications were not mentioned.

i) The manuscript did not discuss the potential application of gene editing technologies (such as CRISPR-Cas9) in the treatment of hypertension, which is an important direction for future genomic research.

j) nsights into specific barriers to translating genomic discoveries into clinical practice.

k) The ethical issues that may arise from genomics research and applications, such as genetic information privacy and genetic discrimination, have not been fully discussed.

---

## [Reviewer Report]

While the present work contributes to this important field, I believe it could be strengthened by considering some recent advances and perspectives in hypertension genomics research.

Please consider the following points:

1. Integration of Multi-Omics Data:

The field is moving towards integrating various ‘omics’ data (genomics, transcriptomics, proteomics, metabolomics) to better understand the pathophysiology of hypertension. Please discuss how this multi-omics approach is providing valuable insights into blood pressure regulation and potential therapeutic targets.

2. Mendelian Randomization Studies

Recent Mendelian randomization studies have been instrumental in identifying causal pathways for hypertension. It would strengthen your manuscript to include several sentences on how these studies are contributing to our understanding of hypertension etiology and potential interventions.

3. Rare Variants and Mendelian Forms of Hypertension:

While the current paper focuses on common variants of primary hypertension, it’s important to also address the role of rare variants and Mendelian forms of hypertension. These can provide valuable insights into blood pressure regulation mechanisms and potential therapeutic targets.

4. Future Directions:

Please discuss the potential of integrating genomic information with clinical data to improve risk prediction and treatment strategies.

Your work significantly contributes to the field of hypertension genomics; however, it may be further strengthened by incorporating recent advances and perspectives. Consider the following points:

1. **Integration of Multi-Omics Data**:

The field is advancing towards integrating various ‘omics’ data, including genomics, transcriptomics, proteomics, and metabolomics, to enhance the understanding of hypertension pathophysiology. Discussing this multi-omics approach could enrich your paper by highlighting how it provides insights into blood pressure regulation and identifies potential therapeutic targets.

2. **Mendelian Randomization Studies**:

Recent Mendelian randomization studies have been pivotal in identifying causal pathways for hypertension. Incorporating a discussion on how these studies advance our understanding of hypertension etiology and inform potential interventions would add substantial value to your manuscript.

3. **Rare Variants and Mendelian Forms of Hypertension**:

While the current focus is on common variants, it is crucial to address the role of rare variants and Mendelian forms of hypertension. This inclusion could offer critical insights into mechanisms of blood pressure regulation and potential therapeutic targets.

4. **Future Directions**:

It is advisable to explore the potential of integrating genomic information with clinical data to enhance risk prediction and inform treatment strategies. This discussion could serve as a forward-looking component of your work.